# A Qualitative Study on Concerns, Needs, and Expectations of Hospital Patients Related to Climate Change: Arguments for a Patient-Centered Adaptation

**DOI:** 10.3390/ijerph18116105

**Published:** 2021-06-05

**Authors:** Benedikt Lenzer, Christina Hoffmann, Peter Hoffmann, Ursula Müller-Werdan, Manuel Rupprecht, Christian Witt, Cornelius Herzig, Uta Liebers

**Affiliations:** 1Institute of Laboratory Medicine, Clinical Chemistry, and Pathobiochemistry, Charité—Universitätsmedizin Berlin, Corporate Member of Freie Universität Berlin and Humboldt-Universität zu Berlin, Augustenburger Platz 1, 13353 Berlin, Germany; christina.hoffmann2@charite.de (C.H.); peter.hoffmann@charite.de (P.H.); 2Department of Geriatrics, Charité—Universitätsmedizin Berlin, Corporate Member of Freie Universität Berlin and Humboldt-Universität zu Berlin, Hindenburgdamm 30, 12200 Berlin, Germany; ursula.mueller-werdan@charite.de; 3Protestant Geriatric Center Berlin, 13347 Berlin, Germany; 4Institute of Physiology, Charité—Universitätsmedizin Berlin, Corporate Member of Freie Universität Berlin and Humboldt-Universität zu Berlin, Charitéplatz 1, 10117 Berlin, Germany; manuel.rupprecht@gmx.net (M.R.); christian.witt@charite.de (C.W.); uta.liebers@charite.de (U.L.); 5Department of Pneumology, Evangelische Lungenklinik Berlin Buch, Lindenberger Weg 27, 13125 Berlin, Germany; cornelius.herzig@jsd.de

**Keywords:** patient participation, patient satisfaction, prevention, treatment outcome, air conditioning, health facility environment, mental health, nursing, education, heat

## Abstract

This study explores the concerns, needs, and expectations of inpatients with the goal to develop a patient-centered climate change adaptation agenda for hospitals. Statements of patients from geriatrics, internal medicine, psychiatry, and surgery (N = 25) of a German tertiary care hospital were analyzed using semi-structured interviews and the framework method. Areas of future adaptation were elaborated in joint discussions with transdisciplinary experts. Concerns included the foresight of severe health problems. The requested adaptations comprised the change to a patient-centered care, infrastructural improvements including air conditioning, and adjustments of the workflows. Guidelines for the behavior of patients and medical services appropriate for the climatic conditions were demanded. The patient-centered agenda for adaptation includes the steps of partnering with patients, reinforcing heat mitigation, better education for patients and medical staff, and adjusting work processes. This is the first study demonstrating that hospital patients are gravely concerned and expect adjustments according to climate change. Since heat is seen as a major risk by interviewees, the fast implementation of published recommendations is crucial. By synthesizing inpatients’ expectations with scientific recommendations, we encourage patient-centered climate change adaptation. This can be the start for further collaboration with patients to create climate change resilient hospitals.

## 1. Introduction

The global trend of a warming climate continued in recent years [1]. The development toward hotter and more extreme weather poses health risks for the human population at large [2] as the relationship between hotter weather, adverse health effects, and hospital admissions is well established [3]. Thus, morbidity and eventually mortality are expected to rise for many vulnerable groups as a consequence of more extreme climatic conditions [4,5].

Loosemore and colleagues describe extreme weather effects such as flooding, storm damages, power outages, or interruption of IT services and water supply on hospitals in Australia and New Zealand [6]. Experiencing such incidents has become a realistic scenario globally, as climate change will lead to extreme weather events in regions with previously temperate climate [7]. These events include more frequent heatwaves and heavy rainfalls [7]. In addition, many hospitals are at risk for excessive heat due to their urban location and the heat-island effect [8]. This results in an increased likelihood of hospital closures, discontinuation of essential medical services, or loss of the coordinating function of hospitals within the health network [6,9].

Beyond these challenges, health systems need to adapt their hospital services to climate change since inpatients make up a particularly vulnerable group. These are often affected by risk factors for heat-related deterioration such as suffering from pulmonary disease or being of very young or old age [10]. Summertime is particularly problematic because “precisely at times when temperatures are high […], hospitals harbor the greatest concentration of vulnerable individuals” [11]. Already, reports on a rising mortality of hospital patients due to detrimental climatic conditions are being published [10,12].

Since hospitals are seen as cornerstones in the adaptation to climate change [8,11], many countries have begun to analyze their healthcare infrastructure and management of severe weather events [13]. While some countries presented comprehensive plans [14] on how to prepare their health services in response only to heat, the WHO published recommendations addressing various climate change dangers [9]. This guidance is comprised of core recommendations central to climate resilience and the transition to sustainability. The recommendations are stressing the importance of a qualified and adequately staffed health workforce, safe access to water, hygiene, and sanitation but also appropriate waste management. It includes guidance on sustainable energy services as well as infrastructure robust to climatic hazards and technologies trimmed for efficient usage.

However, patients are not involved as partners in facilitating the transition process into climate change-resilient health systems. It is noteworthy that no publications on patient views are listed in the sources of adaptation plans. This shortcoming bears the risk of a non-optimal use of resources and of missing adaptation goals, which are important to patients.

The active involvement of patients to consequently empower them is taking on a larger role in health politics, research, and medical care. However, the underlying concepts of patient involvement or patient participation are loosely defined [15,16]. Some health systems, such as the National Health Service in the UK, installed patient participation programs in which patients act as advisors for healthcare providers, managers, or politicians [17]. This engagement fosters joint decision making on all levels of healthcare provision. Patient involvement shares features with the concept of patient-centered care [18]. In patient-centered care, all steps should be directed at the goals of the patients and recipients of care shall maintain a high level of autonomy. Further mainstays are an appreciation of patient views and the development of health literacy. Efforts to strengthen patient involvement and to establish patient-centeredness intend to improve the quality of the delivered care, patient safety, and patient satisfaction [19].

We started this project to lay the foundation for greater patient involvement in the paramount task of preparing hospitals for climate change. Therefore, our aim is to support the adaptation to climate change with evidence on the patient perspective. To start aligning adaptation with the priorities of patients, a needs assessment is required. If patients are involved in this transformation process, the steps of adaptation can be tailored more accurately to their needs.

In this study, we focused on hospital patients. The first objective was to perform a qualitative needs assessment. Our first research question was: What are the concerns and specific needs, and what do inpatients expect to be resolved in the adaptation to climate change?

The second objective was to develop a first adaptation agenda that combines existing adaptation plans with the views of patients. Therefore, our second research question was: Which pressing issues need to be addressed to meet patients’ expectations and to make hospitals fit for climate change?

## 2. Materials and Methods

### 2.1. Data Collection, Setting

To answer the first exploratory research question, we conducted semi-structured interviews with participants from surgery (*n* = 5), geriatric medicine (*n* = 5), internal medicine (*n* = 10), and psychiatry (*n* = 5).

The patient sample was chosen to represent the heterogeneity of hospital patients at risk for negative climate change impacts, such as elderly people [5,20], patients with respiratory diseases [21], patients with higher risk of cardiovascular events [20], surgical patients [22], or children [23]. To better understand the needs and expectations of pediatric patients, also five adult patients with chronic diseases that emerged in childhood were enrolled within internal medicine.

The included wards, only equipped with natural ventilation, are located on three different hospital campuses of the Charité—Universitätsmedizin Berlin. The climate in Berlin is between continental and oceanic.

Semi-structured interviews were audio-recorded by a trained investigator from September 2019 to January 2020. Data were collected at a time when the hot weather of recent years and social movements such as “Fridays For Future” gained broad media coverage. The interview guide consisting of two sections (Appendix A) was developed by the author team and piloted with two patients. The first section deals with the relationship between climate change and health, while the second section contains questions on hospital treatment in light of climate change. The interviewer could ask open-ended follow-up questions for each item of the interview guide.

As a result of the exploratory character of this research endeavor, a purposive sampling strategy was followed. For recruitment, the nurses or ward physicians introduced the study to adult patients with sufficient proficiency in German and the ability to conduct a 20-min interview. The authors did not participate in selecting the patients, and medical treatment was not influenced by this trial. In addition, no incentive was granted to interviewees. The voluntary interview candidates were visited by the interviewer depending on their availability on the ward. After full presentation of the study details, written informed consent was obtained.

This study was approved by the ethics committee of the Charité—Universitätsmedizin Berlin (application number EA4/117/19), and the findings are reported following the Standards for Reporting Qualitative Research [24].

### 2.2. Data Management and Analysis

The data were analyzed with the framework method [25]. This method is a form of content analysis and related to thematic framework analysis. We favored this approach because it can be applied by multidisciplinary teams and has some distinctive characteristics. The framework method follows clear steps facilitating the analysis of multiple transcripts. This approach allows early discussion of a phenomenon, since the synthesis of the findings leads to the outline of a systematical framework. The following paragraphs describe our application of the framework method in this study.

#### 2.2.1. Transcription and Familiarization

The interviews were analyzed after pseudonymization. The author team interpreted the data without being aware of medical history, sex, age, date, and exact location of data recording. The audio files were transcribed verbatim by a team member. Then, the integrity of the dataset was reviewed by the interviewer and two co-authors. The researchers familiarized themselves with the texts by reading the transcripts several times until they felt that they knew the content and the meaning of the patient answers.

#### 2.2.2. Preliminary Coding

Coding was performed using the qualitative data analysis software “QDA Miner LITE” v2.0.7 (Provalis Research, Montreal, QC, Canada). To ensure analytical rigor, each step of the preliminary coding and framework development process was performed by two investigators.

We applied an exploratory and inductive coding technique without having a pre-defined framework. For this purpose, two transcripts of each department were analyzed to generate a code for every sub-theme. More than one code could be applied to the data to facilitate a comprehensive analysis.

#### 2.2.3. Development and Application of the Analytical Framework

The preliminary codes were grouped into themes and sub-themes. These themes and sub-themes were contrasted by two investigators and then standardized and documented in an initial framework. To improve the framework, all 25 transcripts were coded, and themes and sub-themes were iteratively contrasted. If necessary, the themes were refined or expanded. The transcripts and both consecutive versions of the framework were open for discussion to the author team, and the final framework was eventually applied to the 25 interviews by one investigator.

#### 2.2.4. Data Charting with a Framework Matrix and Interpretation

A framework matrix spreadsheet was developed with columns for the themes/sub-themes and a line for each interviewee. This spreadsheet was filled with a summary of the coded paragraphs of each patient. Sample quotes are provided in this manuscript to illustrate patient statements. In the next step, the investigators brought in their professional experience and scientific perspective and jointly interpreted the summaries.

### 2.3. Development of the Agenda

This study is part of a larger research project on the impact of indoor temperature on vulnerable patients (DRKS-ID: DRKS00018692). After finishing the needs assessment, we identified patient priorities in the interview results and searched links to published recommendations and current adaptation plans. Due to the patients’ focus on heat, these plans included the Heat-Health Action Plans of the WHO [26] as well as the Heatwave Plans for England and Germany [27,28]. We also drew on the Lancet Countdown, the WHO guidance on sustainability and resilience, and other relevant publications [8,9]. Based on this evaluation, a preliminary agenda was set that combines patient expectations with measures proposed in the literature. The next step was to invite experts of various fields to review the proposal. To complement the medical expertise of the authors, the experts comprised a public health and urban planning expert (Building Health Lab), a sociologist engaged with societal transformation in the context of climate change (Potsdam Institute for Climate Impact Research), and an expert for architecture and health (European Network Architecture for Health). The authors and external experts jointly elaborated three steps of a patient-centered adaptation agenda for hospitals. In this agenda, we synthesize the needs as seen by patients and related published recommendations to formulate patient-centered priorities for hospital adaptation to climate change.

## 3. Results

### 3.1. Patient Characteristics

Twenty-five adult patients of all age groups were enrolled (21 to 91+ years, 48% female, Table 1).

### 3.2. Needs Assessment

Within the recorded data, two thematic complexes could be distinguished. One complex of grouped themes was related to the concerns of hospital patients, and a second complex assembled needs and expectations. We identified three themes and six corresponding sub-themes in complex 1 and four themes with nine sub-themes in complex 2.

To understand the perspective of the interviewees on the relationship of health and climate change, we asked one yes/no question: 84% of all participants saw a link between health and climate change.

#### 3.2.1. Concerns of Hospital Patients Related to Climate Change

The themes of complex 1 can be summarized as concerns, since the feelings and predictions were predominantly negative. Patients were seriously worried about the impact on their health and their future level of activity. The concerns include the themes “negative predictions and prevalent uncertainty”, “increased burden of disease”, and “consequences for everyday life” (Table 2).


**Theme A: Negative predictions and prevalent uncertainty**


Patients expressed the prediction that the climate will affect the life of patients negatively. For example, it was often stated that the daily life will become more arduous. Envisioned problems included the impairment of medical treatment in hospitals but also the risk of food insecurity that is only indirectly linked to the medical treatment. Statements assigned to both sub-themes revealed that such climate change-related issues cause discomfort.

“[Climate change] will affect me a lot” (G3 (G: Geriatric Medicine, I: Internal Medicine, Psy: Psychiatry, S: Surgery; Numbers indicate different interviewees.)), “If I can’t take the heat, and I have to cope with it and also the bad [weather], the weather plays a constant role” (G2), “Indeed, I believe it will have negative consequences” (Psy4), “It will force me to stay at home, I cannot cope with it, I am very tired, knocked out” (I1).

“Well, treatment in hospitals will become more prolonged, more difficult and invariably more expensive” (G3), “Clearly, we will have food shortages...” (Psy4), “Well, the droughts alone, our diet will have to change dramatically, since we cannot produce the food in this climate” (G3).

It became clear that the developments caused by climate change cause feelings of uncertainty.

“I cannot assess [the situation], yet” (Psy3), “I don’t feel well already” (G2), “I don’t know what climate change looks like” (G4).


**Theme B: Increased burden of disease**


Patients foresaw unfavorable consequences for their health. A recurrent sub-theme in relation to summer weather was the development of fatigue and sleep problems.

“When it’s warm, I feel weak, very tired, as if you performed a long-distance run” (I3), “Well, heatwaves will really exhaust me” (Psy1), “But I think, my sleep deteriorates immediately” (G1).

Excessive heat and longer-lasting heatwaves were seen as the major driver for detrimental climate change consequences. In addition to the mentioning of unspecific health problems, the interviewees also expressed symptoms that can be aligned to disease complexes, e.g., cardiovascular or respiratory disease.

“I have hot flashes; the whole body reacts. I feel really bad” (G2), “If it’s so warm, you have the feeling of dying” (I2), “…It’s like to develop something [e.g., disease symptoms] due to the bad air quality” (I6).

“I have severe cardiovascular problems and such prolonged heatwaves aggravate them” (S5), “You experience more cardiovascular problems” (I10), “Well, it’s hard to breathe” (I5), “Due to the ozone level, the breathing, the lung gets tight” (I8), “Yes, well, blacking out” (Psy1), “Yes, I feel dizzy” (S1).


**Theme C: Consequences for everyday life**


Several interviewees expected their daily activities to be impaired. Participants reported a sharp reduction of mobility on hot days and that they will have to limit their activities due to stronger symptoms. Additionally, they stated that problematic climatic conditions will lead to avoidance behavior. Again, interviewees often referred to warm and dry weather as the cause.

“This will lead to, well, being more at home, since I can’t stand it outdoors” (I1), “I knew, I didn’t go outside” (I9), “You seek the shade, you seek drinks for refreshment, I prefer water without gas, and if I am really thirsty, I bought beer…” (G5).

#### 3.2.2. Needs and Expectations of Hospital Patients Related to Climate Change

In complex 2, participant statements were compiled in the themes “patient-centered services” and “intensified education, staffing, and behavioral aspects” (Table 3). Equipping rooms with air conditioning and improving ventilation was a pressing need for patients of all departments. Therefore, the theme “improved ventilation” was distinguished from “adaptation of the infrastructure and integrating the hospital in the urban environment”.


**Theme A: Patient-centered services**


Participants had the desire for a health-promoting environment that was thoroughly dedicated to their recovery and well-being. From the perspective of a suffering patient, it was stated that they want to feel that every step in their diagnostic workup and treatment furthers their convalescence. Along the same line, they frequently wished that the medical team pays attention to their needs at every encounter. The interviewees requested a culture of caring that is pre-emptively adaptive to the challenges caused by climate change. By that, they mean that hospitals prepare concepts regarding how to meet the needs of patients confronted with adverse climatic conditions.

“That they care… for the well-being of people, that it’s taken care of by them” (G2), “That you can recover somewhere. Sickness is a bad thing, but you do not have to aggravate it” (G2), “Just little things, to make you feel better” (I9), “I think of shelter, a shelter for the weather” (Psy1), “And that you adapt to it, to the conditions and the situation, to facilitate the good and fast recovery” (I6).


**Theme B: Improved ventilation**


The responses show the common understanding that the ventilation needs to be adapted to meet current and future patient needs. Hospitals are perceived as too hot and the possibilities for individual control to achieve thermal comfort in the patient rooms are perceived as insufficient. Participants foremost referred to the installation of air conditioning but also wanted to use the possibilities of natural ventilation (windows). At the same time, patients were aware of the challenges posed by air conditioning such as the prevention of spreading germs, difficulties of ensuring good air quality, and high energy consumption. The air quality itself was seen as a larger issue. The themes raised by respondents included hygiene, adapting the room to disease-specific needs (filtering allergens), and reducing odors.

“These hospitals are way too hot” (G2), “That you have windows to ventilate” (G5), “That [mechanical] ventilation is installed” (G2), “It has to be air conditioning” (I2), “Air conditioning and improved air circulation” (Psy5), “You should install air conditioning” (S1).

“Air conditioning is unhealthy, and secondly, it is very energy-consuming” (G1), “You have to make sure not to spread all the bacteria through the air” (I8), “Air-conditioning is unhandy since it makes the air dry” (I10), “Fresh air in the room” (G1), “Maybe an air purification device… to filter the allergens and odors out of the room” (I3), “Air humidification device” (Psy5).

In addition, we delineated some responses that deal with the subjective perception of the various indoor aspects contributing to thermal comfort such as temperature, air velocity, humidity, and material surface. These expectations have high overlap to the expectations on the interior design in theme C.

“On the one side it’s very cold, the air conditioning, and on the other side: It´s very warm” (S5), “That they adapt to the climatic conditions, that they adjust to it, right now” (I7), “… I found this room to be so cold, I felt really unwell…” (I3).


**Theme C: Adaptation of the infrastructure and integration of the hospital in the urban environment**


Participants shared ideas for the adaptation of the built environment and better equipment. These entailed refitting the building with materials suitable for hotter conditions. Materials with “cooler” surfaces could have multiple advantages such as easier cleaning. In addition, the interior design was addressed, as patients think of a friendly looking environment as beneficial for their well-being.

“It’s not only about medication, it is the whole [room] environment” (I6), “Well, the equipment, so that everything gets better” (I6), “Beds, air conditioning, ventilation, bedding…” (I8), It’s more hygienic and a little bit colder in the summertime…” (I7), “It´s badly organized, in these examination rooms and elsewhere, you don’t have access to oxygen” (I9), “The hospitals should look more pretty, less like a hospital... (I9).

Other statements included important aspects that can help not only to meet the needs of patients but also to transition hospitals to higher sustainability. Patients proposed window shading to keep the rooms cooler and to add more greenery for the purpose of heat mitigation as well as to improve the appearance of the outdoor environment. The ecological footprint of hospitals was brought up as patients want the energy and resource consumption to be reviewed and lowered. As hygiene standards are high and single-use materials are widespread, the recycling/reduction of waste should be increased.

“There should be window shading so that the hospital can be better shaded…” (G5), “Maybe it would be good, real ones, window shadings” (I8), “They should plant greenery on the roofs and plant trees” (G2).

“Do not serve it in plastic [containers] but put the jam in glass jars” (G1), “We would reduce a lot of plastics” (G1), “Avoidance of plastics” (Psy3).

The interviewees were aware of the challenges posed by climate change, which need to be addressed through urban design strategies. Their statements indicate that patients reflect on the hospital surroundings beyond their patient room and expect the health facility to prepare for future challenges.

“To build more recreational areas, for the youngsters and children, I think it’s affecting everybody” (G5), “Maybe spray the area with water, if it’s really hot…. we have a lot of water in Berlin, you would improve the whole city…” (Psy4).


**Theme D: Intensified education, staffing, and behavioral aspects**


Participants felt there to be an urgent need for education on the effects of climate change and suitable behavior. It should be directed at medical staff but also at the individual patient. Counseling on the health effects of climate change, appropriate clothing, physical activity, and fluid intake should be available.

“There should be guidelines for all patients on the ward…” (I7), “Everybody has to adapt, if the climate deteriorates, I have to think for myself, if I go outside if it is 40 °C! (I6).

Patients eventually hope to enhance medical care and to receive more frequent medical consultations in times when the climatic conditions are burdensome by increasing staff numbers.

“To have more staff and time” (G1), “Because the staff is also sweating and is tired” (G5), “With more nursing staff, you could improve everything” (Psy4).

The request to incorporate the climatic conditions in the scheduling of procedures reflects the physical strain that comes with examinations or procedures. Patients were also aware of their own responsibilities to manage exposure to heat stress. Linked to this sub-theme was the notion that long waiting hours should be consistently avoided. The demand for a healthy diet suitable to the weather is linked to this theme.

“Well, examinations should always be adapted to the weather” (I7), “The long waiting times are a really big problem” (G3), “That the staff checks on you more regularly” (Psy5), “Transportation could be better coordinated…”(I7), Nursing staff should take care, due to the medication, that we are not going outdoors if it’s 35°C” (I6), “Remind the patients to drink more” (Psy2), “Adapt the catering” (G1), “Well, to provide enough drinks” (Psy2), “And cold drinks” (S1).

### 3.3. Agenda for a Patient-Centered Adaptation

The immediate need to adapt hospitals and medical care was stressed by 84% of the interviewed inpatients who think their health will be affected by direct and indirect risks of climate change. The reflections of the participating patients show important commonalities with existing frameworks (cf. [8,9]) for adaptation—especially the focus on heat. Our agenda proposes steps that incorporate published recommendations and at the same time focuses on the priorities of patients.

#### 3.3.1. Partnership with Patients in Healthcare

It was clearly expressed in the interviews that the delivered healthcare should not only promote the physical health but also the emotional well-being of the patients. In addition, hospitals should prepare for various future challenges caused by climate change now. Intensified mental health surveillance is needed [29], as participants frequently stated feeling stressed by the changing climate. Therefore, it is necessary to offer specific treatment options for risk groups [26]. Mental health providers educated on issues related to climate change can help patients cope with adverse weather and, thus, avoid reduction of mobility as well as of physical and social activities. Since each patient has unique health priorities and climate change will have different effects depending on each individual’s disease, care teams should respond specifically to their physical needs, too. For example, physical strain can be avoided by rescheduling exercise testing or operations.

To effectively answer the diverse needs and to achieve climate change-adapted medical care, further collaboration and partnership with patients is important. Shared decision-making processes should be integrated on all levels from physician–patient interaction to hospital planning and the setting of regulatory frameworks [17]. In line with the goals of patient-centered care, the study results emphasize that efforts to strengthen partnership with patients should be intensified.

#### 3.3.2. Reinforcement of Heat and Risk Mitigation

High ambient temperatures were a major concern for patients. It is paramount for inpatients to be accommodated in hospitals with convenient indoor temperatures. The participants felt the urge for improvements of ventilation and for refitting of the infrastructure. Beyond that, implementation of urban design approaches to meliorate heat stress and to enhance the appearance of the outdoor environment was supposed.

To be able to adapt to heat and minimize the risks posed by climate change, effective prevention measures need to be implemented. These include improving infrastructure and the built environment [13,30] but also securing essential services such as water, energy, or IT [9,31]. Therefore, the building insulation should be refurbished to avoid overheating [26]. Additionally, natural ventilation should be supplemented with air conditioning to reduce excessive heat. Modern ventilation strategies should include efforts to increase power efficiency, while at the same time, research on the clinical benefits of air conditioning is needed [32]. Moreover, hospitals must also be integrated in early warning systems and in national and local heatwave plans [26,27,28]. These plans should include evidence-based instructions for the hospital management and medical care team [9]. Encouraged by our needs assessment, we propose to accompany these instructions with education material directed at medical professionals and patients, including information on adaptive behavior, medication, and indoor temperature management [9,33,34]. Topics such as the optimal use of natural ventilation, monitoring of the room temperature, and the reduction of indoor heat production should be covered. When critical conditions arise, vulnerable patients should be transferred to cooler, air-conditioned rooms, and the catering should be adjusted.

In accordance with patient expectations to be more resilient to heat stress, hospital planning needs to be linked to urban design strategies. The implementation must start on site with vegetation planting, rain water retention, and passive cooling [35]. Likewise, the creation of cooler indoor public spaces and active travel options safe to use in every weather is needed [36]. Safe access to hospitals must be ensured through the installation of respite areas along the patient journey. These preventive measures help patients stay active regardless of heat exposure or storms. Moreover, hospitals and municipal administrations are required to reduce heat islands by creating more green spaces and to integrate hospitals into cities’ climate change strategies.

#### 3.3.3. Adaptation of Work Processes and Strengthening of Patients’ Coping Capacity

Interviewees recognized risks for the delivery of high-quality medical services in hospitals as well as for their individual condition as a result of climate change. Keeping the health workforce functional while experiencing external stress such as heat or a patient surge was seen as key for the maintenance of medical services. This is a major concern of the WHO [9], too. Creating a positive, holistic hospital experience was stated to be vital to strengthen patients’ coping competencies to restore health and to achieve well-being.

As a consequence, adjusting the processes and the workflows according to the meteorological conditions should be considered. This includes intensified clinical monitoring, a modified drug regimen but also non-medical measures such as adapting clothing, and rescheduling elective procedures to minimize the risks of heat stress [34]. Training of the professional healthcare workers to avoid impairment of services and the best use of “human resources” [9] is vital to mitigate risks for patients. At the same time, flexible work processes can also improve occupational safety, for example by reducing the workload during excessive heat.

To avoid deterioration or recurring hospitalizations, preventive actions are needed to increase the coping capacity of patients. The hospital stay can be a start for education on climate-adapted behavior and suitable physical activities within the hospital but also at home. This opportunity should be used to counsel patients on a sustainable and healthy lifestyle, too. Such trainings should explain the warning signs of heat-related disease deterioration and provide clear instructions on which actions are to be taken. Within the patient-centered approach, the family and the local care network of the patients are to be included, and care should be coordinated before problematic climatic conditions occur. Since behavioral changes are cheaper to modify and faster to implement than infrastructural remodeling [35], this step can already be taken in the near future.

## 4. Discussion

To our knowledge, this is the first study analyzing inpatients’ views on the necessary steps toward climate change adaptation. We uncovered concerns, needs, and expectations of patients vulnerable to environmental hazards. Patients see climate change as a threat to their physical and mental health as well as to the provision of adequate healthcare in hospitals. This project underlines the urgency of adaptation to climate change and reveals an overlap of patient concerns and needs with recommendations of current adaptation frameworks. Participants expressed negative feelings on the prospect of a changing climate and in turn deteriorating health. The expectations summarized in this study are related to workflows in hospitals and the way medical services are provided. Optimal staffing, improved workflows, and a holistic attitude toward care were some of the main requests. Interviewees also urged for a revision and advancement of the infrastructure, foremost the installation of air conditioning.

Furthermore, this exploratory evidence was used to make up for the deficit of insufficient patient participation in climate change adaptation. The results of the interviews were used to propose a first patient-centered agenda for adaptation. Experts from related fields were invited to formulate steps for adaptation jointly with the authors. To emphasize the goals of patients in adaptation, we build on expectations of participants and incorporate existing scientific frameworks. Our concept includes strengthening the partnership with patients in healthcare, reinforcement of heat and risk mitigation, an adaptation of work processes, as well as reinforcing patients’ coping capacity. This patient-centered agenda can accelerate the transition to hospitals resilient to threats of extreme climatic conditions. The results can be used in future research projects or eventually help to implement adaptation measures more efficiently. The interview study can only act as starting point when it comes to consulting patients on adaptation. To review our proposed agenda and at the same time further patient involvement, the formation of focus groups would be the next logical step [37].

### Strengths and Weaknesses

It is a strength of this study that vulnerable adult patients of all ages were included. The interview technique produced answers on a variety of themes that allow broadening the view on adaptation processes beyond a sole medical or technical perspective. Due to conducting the interviews in winter with no heat stress present, some specific aspects of patient care while experiencing heat might not have been evident for the patients.

A bias caused by the selection of participants cannot be ruled out. However, we see the high rate of agreement on themes across the departments and in relation to adaptation programs as a sign for the validity of the registered themes. We also do not see a tendency of social desirability in the answers of interviewees. The review process of the agenda through external experts can be seen as a form of triangulation that supports the validity of the proposed steps. Above all, important aspects were added to the adaptation agenda by the experts, which might help to balance the needs of patients with the priorities of clinicians and researchers.

We are aware that this study with exploratory character has limitations. Since it is a single-center study with a sample of 25 participants in Central Europe, no universal recommendations that fit every setting can be deducted. Patient views on adaptation to climate change might be influenced by specific local features and personal characteristics. In addition, the informal methodology of the agenda development does not allow for a universal application in every adaptation context. Nevertheless, the congruence of the patients’ themes and the problems identified by science show that current adaptation has the correct focus. Much more, the urgency to adapt the hospitals is reinforced through the interviewees. We propose for future studies to interview patients after the discharge from the hospital to avoid the potential bias of not including severely ill patients and expanding this study to other clinical contexts.

## 5. Conclusions

This study shows that inpatients reflect on the various health risks of climate change and view excessive heat as a great hazard. Based on their concerns and needs, patients urgently demand diverse adaptations. Patient views on the problems and risks posed by extreme climatic conditions correspond to the priorities identified by science and stress heat as a major risk. The proposed adaptation agenda combines important goals for patients with current adaptation recommendations.

In conclusion, transdisciplinary efforts are required to create climate change-resilient hospitals in order to ensure patient satisfaction and optimal treatment outcomes. This study might be a starting point for further collaborations with patients in the development of resilient hospitals that strive to meet the goals of patients.

## Figures and Tables

**Table 1 ijerph-18-06105-t001:** Demographic properties of interviewees.

Department	Age Group: *n*	Sex: *n* Female/Male
Geriatric Medicine (*n* = 5) Internal Medicine (*n* = 10) Psychiatry (*n* = 5) Surgery (*n* = 5)	21–30 years: 4 (16%)	12/13 (48%/52%)
	31–40 years: 5 (20%)	
	41–50 years: 2 (8%)	
	51–60 years: 4 (16%)	
	61–70 years: 4 (16%)	
	71–80 years: 2 (8%)	
	81–90 years: 3 (12%)	
	>91 years: 1 (4%)	

**Table 2 ijerph-18-06105-t002:** Overview of themes emerging from the interviews grouped to complex 1.

Complex 1	Concerns of Hospital Patients Related to Climate Change
Theme A	Sub-theme
Negative predictions and prevalent uncertainty	Expectation of negative consequences for individual patients
	Expectation of negative impact on medical care in hospitals and increasing food insecurity
	Expressions of uncertainty
Theme B	Sub-theme
Increased burden of disease	Development/worsening of exhaustion/sleep disorders
	Development/worsening of general symptoms or symptoms related to respiratory and cardiovascular disease
Theme C	Sub-theme
Consequences for everyday life	Avoidance behavior, coping strategies

**Table 3 ijerph-18-06105-t003:** Overview of themes emerging from the interviews grouped to complex 2.

Complex 2	Needs and Expectations of Hospital Patients Related to Climate Change
Theme A	Sub-theme
Patient-centered services	Establishment of “caring” medical services and a “health-promoting” environment
Theme B	Sub-theme
Improved ventilation	Improved ventilation and air conditioning
	Ensuring thermal comfort
Theme C	Sub-theme
Adaptation of the infrastructure and integrating the hospital into the urban environment	Adaptation of infrastructure and built environment
	Increasing sustainability
	Use of urban planning strategies
Theme D	Sub-theme
Intensified education, staffing, and behavioral aspects	Education for staff and patients
	Improvement of staff number and work place ergonomics
	Using the potential of climate change adapted work processes and individual patient behavior

## Data Availability

Data is available from the authors upon request.

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
