# Peer review of "A Qualitative Study on Concerns, Needs, and Expectations of Hospital Patients Related to Climate Change: Arguments for a Patient-Centered Adaptation"

_ijerph, 2021, doi:10.3390/ijerph18116105_

Round 1
Reviewer 1 Report
A qualitative study on concerns, needs and expectations of hospital patients related to climate change: Arguments for a patient-centred adaptation. ijerph-1215303 R1
Thank you for your answer to the reviewers' questions and the thorough review of your paper.
The paper is much improved, but the link between your goal, results, and the process towards developing a patient-centered climate change agenda for hospitals is not clear. Do you think that many hospitals need to adapt their services to meet health problems due to climate change? Do you have any references to this, and can you make this point stronger through more arguments? And what do the agenda process plans look like today? It is important to know if the reader could judge your proposed agenda.
In the method and analysis, write that “the framework method with a positivist approach” (p. 3 and line 141). What do you mean? I do not think that most of the researchers think that this is a positivist approach. Is it important to use this expression?
Your analysis and results. There are still issues with your analysis and report of the results. The analysis does not seem to be complete yet. The number of sub-themes is too many. The main theme could be named differently to mirror the answers better. Your aim was to study the concerns the patient has so the complex/group 1 is unnecessary to name Concerns … since what else could the respondents answer?
I would have main themes that answer that question and name it more informative such as Feelings of insecurity and helplessness. In the text there are so many concepts that you don’t have in the table like helplessness, discomfort etc… Try to condense the transcripts and came up with a name of the theme that mirror the feelings.
I don’t understand the reporting of all the needed adaptation of hospitals that you report on in the result section. We cannot see the link between this and the analysis of the interviews. There is so much more information here that we don’t know where you have taken it from. Which documents is this from? I can see that you have references to needed adaptations. You should not have references in the result section if you don’t have a literature review design. You need to explain the method further on if you report this in the result section.
I would have a section in the introduction of the plans of needed adaptations that you have found in the literature. Then in the discussion I would link this to the patients concern and expressed needs and opinions of the care process. Or I would use a comparative design and use the data from the interviews and from a document analysis or literature review. As it is now the reader get lost in your result presentation.

Author Response
International Journal of Environmental Research and Public Health
– Response to the reviewers
Manuscript ID ijerph-1215303
|
# |
Comment |
Response/ Actions taken |
|
1 |
The link between your goal, results, and the process towards developing a patient-centered climate change agenda for hospitals is not clear.
Do you think that many hospitals need to adapt their services to meet health problems due to climate change? Do you have any references to this, and can you make this point stronger through more arguments?
And what do the agenda process plans look like today? It is important to know if the reader could judge your proposed agenda.
|
The authors would like to thank you for the time you invested to provide this in-depth and profound feedback. Your detailed comments revealed some weaknesses and highlighted specific sections in which the clarity must be improved. We also value your methodological considerations.
The rationale for the study is the insufficient involvement of the target group into the paramount changes which hospitals have to undergo to prepare for the climate crisis. Therefore we performed a qualitative needs assessment and used the results to propose adaptation steps addressing the patient concerns and expectations. The assessment showed that interviewees reflect on diverse topics and that the focus of patients is on heat. Therefore we included heatwave plans and other published recommendations to synthesize these with the needs of patients in the preliminary agenda.
è The rationale/introduction was specified to better explain the links between our aim, our methodology as well as the results of the qualitative analysis and the agenda. Line 90 and following è The agenda development was described in more detail. We expanded the rationale for the inclusion of published recommendations. Paragraph 2.3.
Direct effects of the climate crisis are already witnessed in hospitals. Physicians and administrators are aware of the dangers of e.g. heavier storms, heat spells, interrupted electricity/water/IT-services, surges of acutely admitted patients similar to mass casualty incidents that are related to climate change. Indirect effects include higher disease burden of heat-sensitive conditions, complication of self-management strategies of chronic diseases, occurrence of non-endemic infectious diseases or new stressors affecting mental health. Adaptation strategies place hospitals centrally as “cornerstones in the adaptation”, recommend changes in medical services, and recommend refitting the infrastructure. è We expanded the paragraph on the risks for hospital services/rationale for adaptation with new citations and descriptive examples. Line 47 and following è We added recently published adaptation recommendations Line 65 and following
In addition, we emphasized the necessity to work closely with patients in the adaptation. Crosschecks with the reference lists of all the cited adaptation frameworks and heat wave plans revealed that evidence on patient needs was not included. è We added a short section in the introduction and discussion. Line 74 and 490
|
|
2 |
In the method and analysis, write that “the framework method with a positivist approach” (p. 3 and line 141). What do you mean? I do not think that most of the researchers think that this is a positivist approach. Is it important to use this expression?
|
We followed the Standards for Reporting Qualitative Research (SRQR). It is recommended to include the authors worldview. We understand the reviewer`s notion of the non-agreement of a positivist approach and a qualitative study. Because the authors were mostly educated in positivist thinking, have a biomedical and no social-science training, we thought this was necessary to reveal to judge the analysis. However, we believe that this fact is not important in a public health journal. è This sentence was deleted. |
|
3 |
Your analysis and results. There are still issues with your analysis and report of the results. The analysis does not seem to be complete yet. The number of sub-themes is too many. The main theme could be named differently to mirror the answers better. Your aim was to study the concerns the patient has so the complex/group 1 is unnecessary to name Concerns … since what else could the respondents answer?
|
We realized by your remark that it does not help the understanding if we report the framework in too much detail. è Main-themes were reviewed an re-phrased if necessary, sub-themes were synthesized, the number was reduced and subsequent presentation was changed accordingly.
It is correct that identification of concerns (if prevalent in the data) was an aim of our needs assessment. The open questions in our piloted interview guide did not include specific questions on the concerns (cf. supplement). Much more we wanted to stipulate open thinking and asked in a wider context on the relationship between climate change and health as well as on various aspects of the medical services. Coding was inductive and we had no pre-defined framework for the analysis. Nevertheless, concerns were identified in diverse answers within the whole data-set. Our argument is that our questions did not prompt stating concerns but interviewees reported bundles of troubling developments and negative expectations. Therefore, we believe the summary-term “concerns” to be a good representation of the themes in complex 1.
|
|
4 |
I would have main themes that answer that question and name it more informative such as Feelings of insecurity and helplessness. In the text there are so many concepts that you don’t have in the table like helplessness, discomfort etc… Try to condense the transcripts and came up with a name of the theme that mirror the feelings.
|
Please also refer to our answer above. We condensed the titles of the themes more so that title of the complex and the theme fit better. Since we are aware of the difficulties of condensing data from live interviews (Chi 2009, https://doi.org/10.1207/s15327809jls0603_1) as the epistemology of terms and the meaning of spoken informal language might be incongruent (Csanadi et al., ICLS 2016 Proceedings, https://dx.doi.org/10.22318/icls2016.9), we still use a rather general term for the main-theme and try to describe the precise meaning within the explanatory text. We believe that the re-phrased themes now mirror better the epistemic meaning of the statements.
|
|
|
I don’t understand the reporting of all the needed adaptation of hospitals that you report on in the result section. We cannot see the link between this and the analysis of the interviews. There is so much more information here that we don’t know where you have taken it from. Which documents is this from? I can see that you have references to needed adaptations. You should not have references in the result section if you don’t have a literature review design. You need to explain the method further on if you report this in the result section.
|
Other than in the initial version is the presentation of the agenda within the results a consequence of the preceding review rounds. The request for better alignment of aim and objectives have led to reformulation of our proposal and transfer to the result section. We believe this to be backed by our development process. è à Agenda development and presentation of published recommendations is described now in more detail. è Paragraph 2.3. è è à To clarify the connection to the interview analysis, we re-structured the arguments in the agenda and sum up in short the opinion of patients and formulate synthesized recommendations on the basis of published recommendations. |
|
6 |
I would have a section in the introduction of the plans of needed adaptations that you have found in the literature. Then in the discussion I would link this to the patients concern and expressed needs and opinions of the care process. Or I would use a comparative design and use the data from the interviews and from a document analysis or literature review. As it is now the reader get lost in your result presentation.
|
Please also refer to the answers above.
è We inserted published adaptation recommendations in the introduction Line 65 and following
è We restructured our proposed steps for the agenda to better delineate concern and needs from recommendations from literature.
|
Please see the attachment

Reviewer 2 Report
The manuscript has undergone a major improvement in terms of clarity and rigor. By clearly changing the wording in parts 3.3.1 and 3.3.2, the work has more clearly highlighted one of the areas, the significance of climate change and how we should adapt to these changes. The discussion is also more clearly written and includes aspects that deepened the analysis.
Author Response
Thank you
Reviewer 3 Report
The authors clearly improved the manuscript, therefore, in my opinion, it is now susceptible for publication
Author Response
Thank you
Round 2
Reviewer 1 Report
Thanks for the revisions and clarifications. I would go through the paper again for language check and for reference presentations
This manuscript is a resubmission of an earlier submission. The following is a list of the peer review reports and author responses from that submission.
Round 1
Reviewer 1 Report
Comments on the manuscript “A qualitative study on concerns, needs and expectations of hospital patients related to climate change: Arguments for a patient-centred adaptation”
Manuscript: ijerph-1088596
Glad to read and comment on your manuscript about the patient's participation in adapting hospitals to the impact of climate change. This is an important issue and will become even more important in the future. As you say, hospital buildings as a system are at high risk to be affected by heat waves etc caused by climate changes and hospitals harbour the greatest concentration of vulnerable individuals. At the same time, hospitals are one of the worst carbon dioxide emitters. One important focus in your study is how to involve the patient into climate change-resilient hospital changes.
In your title you write that this is a qualitative study but the method you used is purely quantitative. “A qualitative study on concerns, needs and expectations of hospital patients related to climate change”.
In the abstract you write that the aim is to fostering patient participation in the adaptation of hospitals to climate change impacts like heat and other severe weather events. Have you study this how to foster patient participation?
In the abstract you mention “framework method” what is that? Do you mean thematic framework method?
The main problem with your introduction is that you do not explain what your paper adds to the field. Do you want to study how the patient can be more involved in adaptation of hospital designs? Or what their concern are about climate changes? What are your research questions?
The method section needs to be clarified. You need to describe each step in your method. According to the author you refer to the framework method includes five steps; familiarization; identifying a thematic framework; indexing; charting; and mapping and interpretation. You use other concepts like topics etc… that confuse the clarity.
How did you recruit the patients?
The section when you describe the researchers background can you move to a method discussion section or delete. Within the recorded data, two thematic complexes could be distinguished.
The result section is unclear. I got lost in your result presentation. I think it is confusing when you present the data as you do with the dot tables. In a qualitative analyze you synthesize the participants expressions. In addition, where they have been patients are less important. This was not your research question either.
You need to show us in a table without dots – what is the main themes and sub-themes? In the result you should express what the participants answered, like The participants expressed that …. The participants said that…
I don’t see the link between the questions that you raised to the participants and the answer you got. How could one theme be “Link between health and climate change (yes/no-question)” that seems to be a question.
The analysis needs to be developed.
In the discussion your results should be discussed related to other studies and your interpretation of what the results could mean be presented.
Author Response
Please see the attachment
International Journal of Environmental Research and Public Health
– Response to the reviewers
Manuscript ID ijerph-1088596
|
# |
Comment |
Response/ Actions taken |
|
1 |
Rev_1: In your title you write that this is a qualitative study but the method you used is purely quantitative.
|
Thank you very much for your effort and the extensive feedback. The points raised are of great value and we perceive the in-depth guidance as a chance to improve the paper`s clarity and quality. Yours and your colleagues constructive feedback has led to a restructured manuscript and an improved proposal for a patient-centered adaptation.
We believe that we caused confusion regarding our qualitative approach by the semiquantitative display of participant`s answers in our tables. Also, we did present all answers in the tables related to the medical department in which the patients were treated. This could cause the impression that we would contrast answers between departments. However, it was meant to preserve information potentially valuable for upcoming research questions. è To improve clarity, we now present the results of our analysis without focusing on the department. è As a consequence, we now restructured the results section. We further explain our rationale in the next answers. è We accentuated some paragraphs to clarify that this study comes with classic features of qualitative methods like - semi-structured interviews, a typical instrument of qualitative research - We describe the phenomenon of patient views on problems caused by climate change and the necessary adaptations - The analysis of the interviews is inductive and exploratory - The results can generate new hypothesis and can create a better understanding for the concerns and needs of patients
|
|
2 |
Rev_1: In the abstract you write that the aim is to fostering patient participation in the adaptation of hospitals to climate change impacts like heat and other severe weather events. Have you study this how to foster patient participation?
|
We agree that we must specify the aim of the study better. It was not our intention to analyze in which way patients are integrated best into the process of adapting hospitals to climate change. We strived for a first exploration of the problems that worry patients confronted with climate change and to get an impression about their needs for adaptation from their own perspective. We think it is critical to learn from the very start of the adaptation processes from the target group to start a bi-directional communication process and to identify priorities that need to be challenged first.
|
|
3 |
Rev_1: In the abstract you mention “framework method” what is that? Do you mean thematic framework method?
|
Correct, the framework method, as outlined by Gale et al. (https://doi.org/10.1186/1471-2288-13-117), is part of the various thematic framework methods. This method is used for the content analysis of qualitative data. The framework method shows some distinctive features: It entails clear steps but focusses, in our understanding, stronger on the inductive coding. We see an advantage in the structured approach since larger numbers of transcripts can be accessed. According to Gale et al., only after the coding, the identification of the themes or the building of the framework follows, respectively. Within that framework the data is summarized and synthesized.
è We added a detailed explanation of our analysis approach: Paragraph 2.2. and following |
|
4 |
Rev_1: The main problem with your introduction is that you do not explain what your paper adds to the field. Do you want to study how the patient can be more involved in adaptation of hospital designs? Or what their concern are about climate changes? What are your research questions?
|
è We added a better explanation of the problem at hand. Line 59-62, 75-78 è Research questions were added and the objectives were substantiated. Line 79-86
|
|
5
|
Rev_1: The method section needs to be clarified. You need to describe each step in your method. According to the author you refer to the framework method includes five steps; familiarization; identifying a thematic framework; indexing; charting; and mapping and interpretation. You use other concepts like topics etc… that confuse the clarity.
|
è We harmonized the terminology and use now “themes” and “sub-themes” to describe the levels of the framework. è We expanded and clarified each step of our approach. Paragraph 2.2. and following |
|
6 |
Rev_1: How did you recruit the patients?
|
è We rewrote the paragraph on participant recruitment and reported the process in detail. Line 115-125 |
|
7 |
Rev_1: The section when you describe the researchers background can you move to a method discussion section or delete. Within the recorded data, two thematic complexes could be distinguished.
|
è The paragraph was shortened and added to the report in the method section on the development of the agenda. |
|
8 |
Rev_1: The result section is unclear. I got lost in your result presentation. I think it is confusing when you present the data as you do with the dot tables. In a qualitative analyze you synthesize the participants expressions. In addition, where they have been patients are less important. This was not your research question either.
|
We can understand your criticism. è The result tables were restructured. The dots were deleted.
|
|
9 |
Rev_1: You need to show us in a table without dots – what is the main themes and sub-themes? In the result you should express what the participants answered, like The participants expressed that …. The participants said that…
|
We developed the analysis and presentation of the results further. è We reduced the number of tables and now display only two tables with themes and sub-themes to give an overview on our framework. è We expanded the representation of the synthesis of patient answers and embedded the sample quotes more naturally in the text. è We hope that the results now allow a more fluent understanding of the insights provided by the patients.
|
|
10 |
Rev_1: I don’t see the link between the questions that you raised to the participants and the answer you got. How could one theme be “Link between health and climate change (yes/no-question)” that seems to be a question.
|
We report the results of the analysis of the data we collected in semi-structured interviews with open-ended questions. Thanks to the extensive feedback of the reviewers è we clarified our method and the presentation. We hope that this manuscript is now more accessible. è we clarified and explained the yes/no question in line 193-195 |
|
11 |
Rev_1: The analysis needs to be developed.
|
è After restructuring the tables, the results and analysis were rewritten in parts/expanded and we also restructured this part. è We aligned the results according to the research questions
|
|
12 |
Rev_1: In the discussion your results should be discussed related to other studies and your interpretation of what the results could mean be presented.
|
è After scrutinizing the comments of the reviewers, we chose the approach to develop the agenda for adaptation further. In this section, we discuss the patient`s insights combined with existing literature and elaborate the resulting implications. è Paragraph 3.3 |

Reviewer 2 Report
Interesting and significant contribution to the field of research that works with questions about healthcare's adaptation to climate change. Adaptation is central to studying whether we should be able to meet the effects of climate change. The following are some comments and reflections:
Abstract
p 1, line 21-22, this aim and the aim at p 2 63-65 are not consistent, which makes it confusing for the reader.
Introduction
Throughout, there is no theoretical basis for the concept of "patient-centered care". Linked to the title where this concept is clearly emphasized, there is no in-depth study of what patient-centered is and means in healthcare. Why this perspective and what is the benefit of healthcare being patient-centered? An in-depth study of the concept patient-centered in the background and then a connection to the concept in the discussion where the result is reflected against the theory should be a good contribution to develop the article.
Method
p 2, lines 94-96. As a reader, I would like to have a few lines about what this framework is. What makes it different from other methods?
Results
The tables are difficult to read, which may be explained by the fact that they are not completely proofread. For example, Table 1 needs clearer lines between the different parts, as it is now difficult to see which age group belongs to which department.
Discussion
See the comment under Introduction. The article would benefit from a clearer theoretical feedback to person-centered care and why this would be a benefit regarding the adaptation of the health sector.
p 9, 259. Develop this statement. The sentence "Therefore, responses to climate change should include need-adapted medical services" is left as it is, now open and as a reader I want to know what solutions are proposed?
Author Response
International Journal of Environmental Research and Public Health
– Response to the reviewers
Manuscript ID ijerph-1088596
|
# |
Comment |
Response/ Actions taken |
|
13 |
Rev_2: p 1, line 21-22, this aim and the aim at p 2 63-65 are not consistent, which makes it confusing for the reader.
|
We are grateful for your effort and the feedback. The points raised are of great value and we perceive the in-depth guidance as a chance to improve the paper. Eventually, we want to foster the clarity and the quality of our manuscript. The constructive feedback of you and your colleagues led to a, in large parts, restructured manuscript and the further development of our proposal for a patient-centered adaptation.
è Research questions were added and the objectives were substantiated. è The aim in the abstract was re-phrased for clarification. |
|
14 |
Rev_2: Introduction Throughout, there is no theoretical basis for the concept of "patient-centered care". Linked to the title where this concept is clearly emphasized, there is no in-depth study of what patient-centered is and means in healthcare. Why this perspective and what is the benefit of healthcare being patient-centered? An in-depth study of the concept patient-centered in the background and then a connection to the concept in the discussion where the result is reflected against the theory should be a good contribution to develop the article.
|
è We added a paragraph detailing the overlapping concepts of patient involvement/participation and patient-centered care in the introduction. In the next step, we outline our approach how we want to bolster the adaptation through integration of patient views. Line 63-74
è We pick up the theme of patient participation and patient-centered care as we discuss the implications and next steps in relation to the agenda in the discussion. Line 396-401, 458-460, 476-485.
|
|
15 |
Rev_2: Method p 2, lines 94-96. As a reader, I would like to have a few lines about what this framework is. What makes it different from other methods?
|
Other than the content analysis technique following Mayring or the grounded theory method, the framework method is suitable for multidisciplinary teams in which not every member is trained in social sciences or some members have less experience with qualitative data. Since the necessary steps are clear and logical and the analysis phase is based on a structured matrix, it allows joint discussion and input from various expertise fields. We favored this structured approach since we conducted interviews with lay-persons and not experts in the field. Therefore, we expected the data not to be as rich as when we would interview e.g. public health researchers. Another advantage compared to other methods is that with the framework method many cases (patients) can be integrated at the same time. è We added a detailed explanation of our analysis approach. Paragraph 2.2. è We expanded and clarified each step of our approach. Paragraph 2.2.1. – 2.2.4.
|
|
16 |
Rev_2: Results The tables are difficult to read, which may be explained by the fact that they are not completely proofread. For example, Table 1 needs clearer lines between the different parts, as it is now difficult to see which age group belongs to which department.
|
è All tables were revised, table margins and lines were placed. We hope that the tables are now easier to read. è Table structure (dots) was changed according to the recommendation of the other reviewers |
|
17 |
Rev_2: Discussion See the comment under Introduction. The article would benefit from a clearer theoretical feedback to person-centered care and why this would be a benefit regarding the adaptation of the health sector.
|
We believe that the new structure of the manuscript and the conceptualization of the agenda now provide a more general and accessible presentation. We aimed at highlighting the benefits of patient involvement in adaptation and care through the description of the agenda and the discussion. |
|
18 |
Rev_2: p 9, 259. Develop this statement. The sentence "Therefore, responses to climate change should include need-adapted medical services" is left as it is, now open and as a reader I want to know what solutions are proposed?
|
We included a whole section (3.3. agenda) with a proposal on solutions for problems that have been raised.
|
Please see the attachment

Reviewer 3 Report
This manuscript provides us information about needs and expectations of hospital patients related to climate change. The topic is interesting and the study design and the study procedure are very clear. However the study has a very small sample, so the authors must be careful when interpreting the results. In fact, some conclusions are very preliminary. I recommend that the authors increase the sample.
I would like to make several suggestions for revision:
Abstract
Line 23-24: “ Semi-structured interviews with 25 patients of geriatrics,…”- In the abstract the authors wrote that they interviewed 25 patients, but in the results they referred 25 patients of both sexes. The authors need to clarify this aspect
Introduction
Line 41 – please provide a reference in the end of “….the human population at large.”
Line 63-65: “The first objective of this qualitative study was to assess the needs and explore the concerns and expectations of hospital patients in the context of climate change. Based on these findings, the second objective was to inform the next steps of adaptation.” How the authors are going to perform the second objective? The authors need to clarify this topic.
Material and methods
Line 73-“ [7], [10], [24-25])”. When we have several references, they must be referred in the text as well [7, 10, 24-25]. Please revised all the text
Line 96– Please spell out “QDA Mine” for de readers to understand the name
Results
Table 2- Provide a line between the themes of the complex 1 and the complex 2, for not misinterpretation
Table 3- the authors wrote in sample quotes G2, G4, G2, Psy1… Despite they wrote a note in the end of the table to explain what the meaning of the terminology G, Psy, … they need to clarify the meaning of the numbers, in the other words, what it the meaning of G2, Psy1, … this is not clear for the reader.
Author Response
International Journal of Environmental Research and Public Health
– Response to the reviewers
Manuscript ID ijerph-1088596
|
# |
Comment |
Response/ Actions taken |
|
18 |
Rev_3:
However the study has a very small sample, so the authors must be careful when interpreting the results. In fact, some conclusions are very preliminary. I recommend that the authors increase the sample. |
Thank you very much for your kind words and the valuable comments on our manuscript. We worked hard to resolve the open issues and hope to meet your expectations.
Unfortunately it was not possible to include more patients at this stage of project. à Therefore, we reinforced our statements on the exploratory character of the study and the limited transferability. Line 488-499 & 511-514.
|
|
19 |
Rev_3: Abstract Line 23-24: “Semi-structured interviews with 25 patients of geriatrics,…”- In the abstract the authors wrote that they interviewed 25 patients, but in the results they referred 25 patients of both sexes. The authors need to clarify this aspect
|
è We included 10 patients in internal medicine and 5 patients of every other department. We clarified this aspect within line 90-92. è The abstract was revised. è After restructuring of the text, Table 1 now clarifies this information.
|
|
20 |
Rev_3: Introduction Line 41 – please provide a reference in the end of “….the human population at large.”
|
è We added the reference „McMichael, A.; Montgomery, H.; Costello, A. Health risks, present and future, from global climate change. BMJ 2012, doi: BMJ 2012;344:e1359“.
|
|
21 |
Rev_3: Line 63-65: “The first objective of this qualitative study was to assess the needs and explore the concerns and expectations of hospital patients in the context of climate change. Based on these findings, the second objective was to inform the next steps of adaptation.” How the authors are going to perform the second objective? The authors need to clarify this topic.
|
è We clarified the objectives and research-questions in the introduction-section. è After scrutinizing the comments of the reviewers, we chose the approach to develop the agenda for adaptation further. In this section, we discuss the patient`s insights combined with existing literature and elaborate the resulting implications. Paragraph 3.3 |
|
22 |
Rev_3: Material and methods Line 73-“ [7], [10], [24-25])”. When we have several references, they must be referred in the text as well [7, 10, 24-25]. Please revised all the text
|
We have revised the entire text and provided in-text references where appropriate, e.g. Line 93-96 “…such as elderly people [5, 18], patients with respiratory diseases [19], patients with higher risk of cardiovascular events [18], surgical patients [20] or children [21].“
|
|
23 |
Rev_3: Line 96– Please spell out “QDA Mine” for de readers to understand the name |
QDA Miner is the proper name of this software similar to SPSS. è To highlight this, we spelled out and placed the name in quotes. Line 145-146 |
|
24 |
Rev_3: Results Table 2- Provide a line between the themes of the complex 1 and the complex 2, for not misinterpretation
|
è Following the requests of your fellow reviewer and your recommendations for restructuring the paper, the table was deleted.
|
|
25 |
Rev_3: Table 3- the authors wrote in sample quotes G2, G4, G2, Psy1… Despite they wrote a note in the end of the table to explain what the meaning of the terminology G, Psy, … they need to clarify the meaning of the numbers, in the other words, what it the meaning of G2, Psy1, … this is not clear for the reader.
|
We added the explanation in a footnote „G: Geriatric Medicine, I: Internal Medicine, I_Ped: Internal Medicine (Pediatrics), Psy: Psychiatry, S: Surgery; Numbers indicate different interviewees.“ to clarify the meaning of these numbers. Page 6_footnote
|
Please see attachment

Round 2
Reviewer 3 Report
The authors clearly improved the manuscript, therefore, in my opinion, it is now susceptible for publication